# SARS-CoV-2 spike E156G/Δ157-158 mutations contribute to increased infectivity and immune escape

Tarun Mishra[1], Rishikesh Dalavi[1], Garima Joshi[2] , Atul Kumar[3,4], Pankaj Pandey[4], Sanjeev Shukla[4,5] , Ram K Mishra[2,4], Ajit Chande[1,4]

Breakthrough infections by emerging SARS-CoV-2 variants raise significant concerns. Here, we sequence-characterized the spike gene from breakthrough infections that corresponded to B.1.617 sublineage. Delineating the functional impact of spike mutations revealed that N-terminal domain (NTD)-specific E156G/Δ157-158 contributed to increased infectivity and reduced sensitivity to vaccine-induced antibodies. A six-nucleotide deletion (467–472) in the spike-coding region introduced this change in the NTD. We confirmed the presence of E156G/Δ157-158 from cases concurrently screened, in addition to other circulating spike (S1) mutations such as T19R, T95I, L452R, E484Q, and D614G. Notably, E156G/Δ157-158 was present in more than 90% of the sequences reported from the USA and UK in October 2021. The spike-pseudotyped viruses bearing a combination of E156G/Δ157-158 and L452R exhibited higher infectivity and reduced sensitivity to neutralization. Notwithstanding, the post-recovery plasma robustly neutralized viral particles bearing the mutant spike. When the spike harbored E156G/Δ157-158 along with L452R and E484Q, increased cell-to-cell fusion was also observed, suggesting a combinatorial effect of these mutations. Our study underscores the importance of non-RBD changes in determining infectivity and immune escape.

## Introduction

During the second wave of the COVID-19 pandemic, there was a substantial rise in the number of cases in India, reaching more than 400,000 cases per day (WHO, 2019). The extent of spread was attributed to fitness-conferring mutations in the parental lineage B.1.617, leading to the emergence of sublineages such as B.1.617.1, B.1.617.2, and B.1.617.3 of SARS-CoV-2 (Rambaut et al, 2020). This emergence of variants coincided with the vaccination drive, prioritized for the frontline workers, older population with subsequent rollouts in high-risk groups, and young adults. While the front-line workers mostly received both doses of ChAdOx1 nCoV-19 (Covishield in India) by March 2021, highly transmissible variants such as delta (B.1.617.2) displayed the ability to cause breakthrough infections (Ujjainiya et al, 2021 Preprint). Our surveillance analysis also identified cases that were classified as vaccine breakthrough infections, and it became pertinent to understand the potential of the pathogen targeting vaccinated individuals.

SARS-CoV-2 entry is mediated by the interaction of its spike (S) glycoprotein on the virions with the human angiotensin-converting enzyme 2 (ACE2) receptor (Hoffmann et al, 2020). The spike protein from B.1.617 lineage harboring L452R and E484Q mutations was reported to have contributed to the pathogenicity (Ferreira et al, 2021; Rajah et al, 2021). These mutations are present in the critical receptor-binding domain (RBD), a target for neutralizing antibodies. Indeed, recent reports demonstrated diminished sensitivity of spike PVs bearing L452R and E484Q to BNT162b2 mRNA vaccine–elicited antibodies but a lack of synergy between these two mutations in conferring the resistance (Ferreira et al, 2021; Liu et al, 2021a).

Although spike-focused vaccines designed based on the seeding variants have been shown to prevent symptomatic disease effectively (Polack et al, 2020; Voysey et al, 2021), the ability of the emerging variants to breakthrough vaccine-elicited host defense was attributed to escape mutations in the spike (Kang et al, 2021; Peacock et al, 2021 Preprint). Despite the restricted tropism to ACE2-expressing cells, the spike appears tolerant to mutations that confer the ability to escape humoral immunity and, by extension, the resistance to antibody treatments (Gupta et al, 2021). Therefore, to understand the antigenic alterations in the spike protein underlying reduced vaccine effectiveness in breakthrough infections, we cloned and sequence-characterized spike genes from RT-PCR–positive cases that were either vaccinated or did not receive the vaccine. The spike nucleotide sequence analysis revealed a series of shared mutations that we functionally characterized using reporter pseudoviruses (PVs) (Mishra et al, 2021). We observed that

[1]Molecular Virology Laboratory, Department of Biological Sciences, Indian Institute of Science Education and Research, Bhopal, India  [2]Sumo and Nuclear Pore Biology Group, Department of Biological Sciences, Indian Institute of Science Education and Research, Bhopal, India  [3]Structural Biology Laboratory, Department of Biological Sciences, Indian Institute of Science Education and Research, Bhopal, India  [4]COVID-19 Testing Centre, Indian Institute of Science Education and Research, Bhopal, India  [5]Epigenetics and RNA Processing Laboratory, Department of Biological Sciences, Indian Institute of Science Education and Research, Bhopal, India

Correspondence: ajitg@iiserb.ac.in

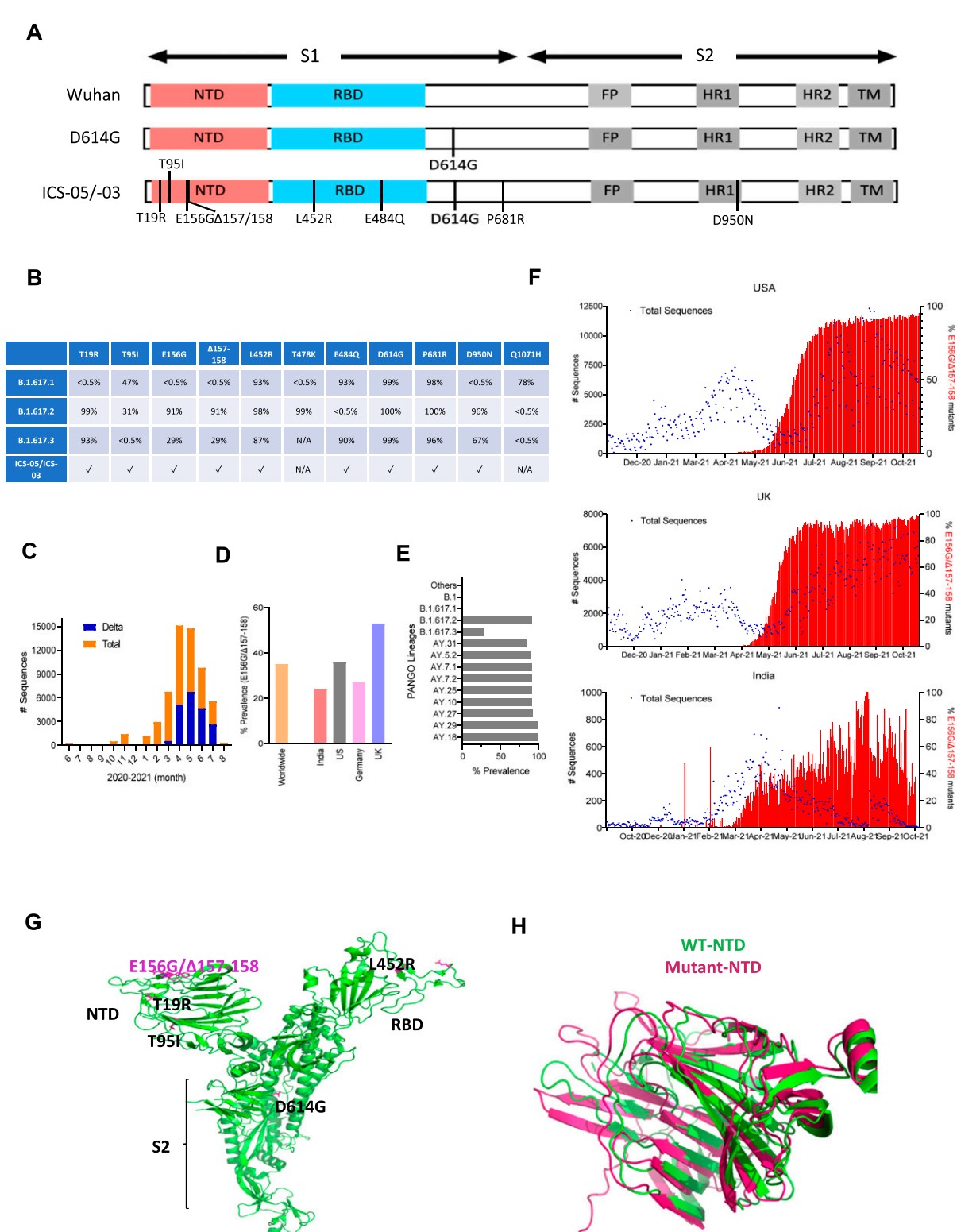

**Figure 1. Sequence characterization, geographical prevalence, and structural implications of spike mutations.**
**(A)** Schematics of the spike protein representing the mutations in the S1 and S2 domains. Different regions of spike proteins are indicated: NTD (N-terminal domain), receptor-binding domain, FP (fusion peptide) HR1 and 2 (heat repeat 1 and 2), and TM (transmembrane region). **(B)** The table represents the frequency of various mutations found in the SARS-CoV-2 sequences submitted on GISIAD till October 2021. **(C)** Bar graph indicates the frequency of the Delta variant during the second wave in India. The orange color shows the total number of SARS-CoV-2 sequences submitted each month (denoted on the X-axis), and the blue represents the number of sequences among total sequences submitted between the period of June 2020–August 2021. The data were obtained from the GISIAD SARS-CoV-2 database

amino acid changes in the spike N-terminal domain (NTD), which are also shared by emerging variants worldwide, contribute to increased infectivity, resistance to vaccine-elicited polyclonal antibodies, and cell-to-cell fusion.

# Results

## Spike gene from breakthrough infections share a recurrent six-nucleotide deletion

During our surveillance study, we identified previously uninfected, ChAdOx1 nCoV-19 (Covishield) fully vaccinated cases that were infected ~50 d after the second dose. In an effort to map the associated spike mutations that plausibly enabled the virus to break through the host defense, a full-length spike gene was PCR-amplified using reverse-transcribed cDNA as a template. Sanger sequencing identified substitutions that were analyzed to comprehend the origin of the spike protein variant from a breakthrough infection (termed hereafter ICS-05 and ICS-03). Comparison of ICS-05 and ICS-03 spike sequence with the Wuhan isolate spike sequence revealed a total of eight changes; three were in NTD, four in RBD, and one on the S2 portion of ICS-05 spike (Figs 1A and S1). Interestingly, we observed a six-nucleotide deletion that resulted in the loss of two amino acids at 157 and 158 positions and a change of glutamic acid at 156 positions to glycine (E156G/Δ157-158) (Fig S1). We found this deletion in five of the total seven spike sequences obtained from the RT-PCR–positive cases (Fig S1). Of these five cases, two (ICS-05 and ICS-03) were fully vaccinated, closely associating this deletion with breakthrough infections. When the ICS-05/03 spike mutational profile was compared with other spike sequences from the B1 lineages (Elbe & Buckland-Merrett, 2017), they both corresponded to the B.1.617.3 lineage (Fig 1B). Notably, the delta variant (B.1.617.2) that dominated the second wave in the country and caused 25.3% of breakthrough infections (Ujjainiya et al, 2021 *Preprint*) shares all spike mutations with B.1.617.3 except T478K (Fig 1B and C). The E156G/Δ157–158 mutation, first detected on 7 August 2020, subsequently became 35% prevalent worldwide (Fig 1D), and by October 2021, it was found in more than 90% of reported sequences from the USA and UK, with a downward trend from India (Fig 1F). While the E156G/Δ157-158 mutation was underrepresented in parental B.1 lineage, it was detected with high frequency in at least 157 countries in B.1.617.2 and B.1.617.3 lineage and was found in multiple PANGO lineages, including the AY lineage (Elbe & Buckland-Merrett, 2017; https://outbreak.info/) (Fig 1E). Given the higher prevalence (Fig 1D and F), we hypothesized the virological significance of these non-RBD mutations E156G/Δ157-158.

We first examined the spike protein in the structural context of E156G/Δ157-158 for clues regarding the alteration of epitopes. Interestingly, mapping of E156G/Δ157-158 on the structure of wild-type spike protein implies that the mutated region is surface exposed, which might be a good target for antibodies (Fig 1G). Furthermore, to understand the effect of these mutations on spike protein structure, we predicted the structure of NTD-bearing E156G/Δ157-158 using the AlphaFold (Tunyasuvunakool et al, 2021). The resultant model of a mutant spike protein does not show any significant changes in the NTD of the spike protein (Fig 1H), suggesting resistance to neutralizing antibodies may not be attributed to structural changes.

## E156G/Δ157-158 contributed to attenuated neutralization susceptibility and increased infectivity

To appraise the exact potential of the NTD bearing E156G/Δ157-158 and the changes found in the region important for receptor binding, we introduced indicated mutations on the parental D614G (B.1) spike gene by site-directed mutagenesis (Fig 2A). Next, the impact of these spike mutations on the infectivity of PV was assessed following the earlier reports (Ferreira et al, 2021; Mishra et al, 2021). The lentiviral spike PVs carried a luciferase gene, and the values were represented after normalizing to the milli units of reverse transcriptase (RT mU). In agreement with previous findings (Ferreira et al, 2021; Rajah et al, 2021), whereas the RBD-specific mutation E484Q did not significantly confer infectivity advantage to the spike particles, the L452R mutation increased the infectivity more than twofold in these conditions (Fig 2B).

Interestingly, the spike E156G/Δ157-158 mutation (present in the NTD) conferred infectivity advantage almost equal to that of L452R (present in the RBD) (P-value of <0.0001 from one-way ANOVA). In comparison, the remaining NTD-specific mutations examined (T19R, T95I, and T19R/T95I) did not significantly confer infectivity advantage (Fig 2B). Western blotting from the producer cell lysates and purified virions indicated that all these spike protein mutants were expressed, and there was no noticeable effect on the processing of spike or its virion incorporation (Figs 2C and S2).

Next, we examined the susceptibility to neutralization of indicated spike PVs to vaccine-elicited plasma polyclonal antibodies from the test-negative individuals who received the vaccine doses on the same day with a 1-mo time interval between two doses. With the D614G as a reference, the $NT_{50}$ values (see the Materials and Methods section for details for $NT_{50}$) obtained showed that spike PV carrying the E156G/Δ157-158 mutation was 4.85-fold less susceptible (P-value <0.01 from Wilcoxon signed-rank test) to vaccine-elicited polyclonal antibodies (Fig 2D), indicating the role of this mutation in escaping the vaccine-elicited antiviral immunity in addition to promoting virion infectivity. The other indicated mutations, however, did not confer noticeable resistance in these conditions except the L452R mutant–bearing PV that required a 2.36-fold higher plasma for neutralization (P-values < 0.01 from Wilcoxon signed-rank test), which is consistent with the previous findings (Ferreira et al, 2021) (Fig 2D). Altogether, these results suggest the contribution of the NTD-specific mutations, in addition

(https://outbreak.info/). **(D)** The prevalence of E156G/Δ157-158 mutation in the indicated countries and worldwide. **(E)** Occurrence of E156G/Δ157-158 in the PANGO lineages (https://cov-lineages.org/index.html). **(F)** The numbers of sequences carrying E156G/Δ157-158 were reported by indicated countries. The left Y-axis represents the total number (#) of SARS-CoV-2 genome sequences (blue dots), whereas the right Y-axis denotes the percentage of E156G/Δ157-158 occurrence (red bars). The X-axis represents the months of sequence submission. **(G)** Mutations found in the spike ICS-05 are shown in the sticks (magenta) in the spike protomer (green, PDB ID: 7DF3). **(H)** Superimposition of WT NTD (green, PDB ID: 7DF3) and E156G/Δ157-158 NTD (magenta) of the spike protein.

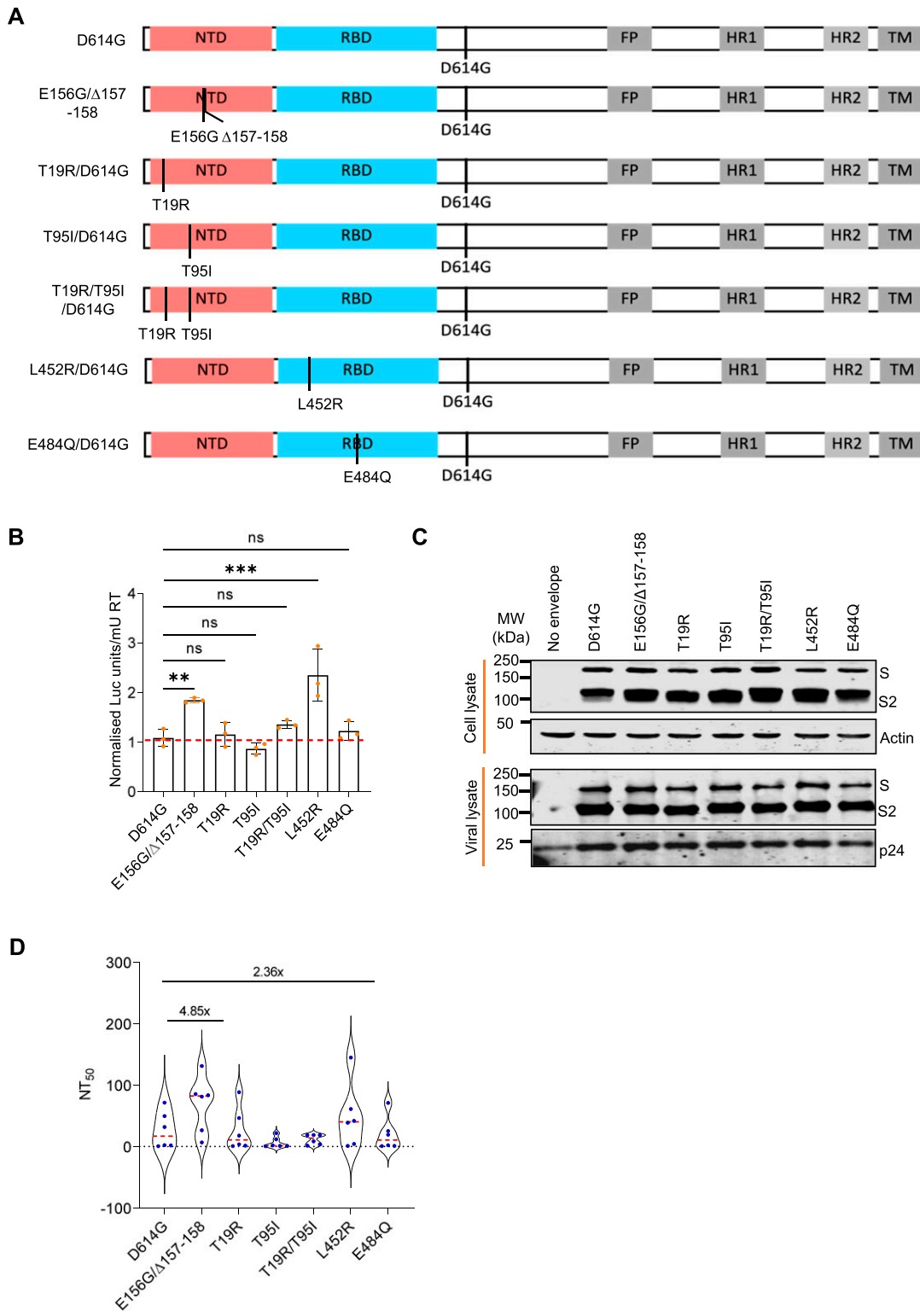

**Figure 2. Infectivity and neutralization of spike pseudoparticles.**
**(A)** Schematics of the spike mutants generated to study the effect of ICS-05/03–specific mutations. Amino acid positions are represented with respect to the Wuhan HU-1 sequence (NC_045512). **(B)** Infectivity profiles of the indicated spike mutant–pseudotyped lentiviruses. The infectivity was normalized to the D614G-pseudotyped lentiviral particles. The data represent the mean of three replicates, and the significance was measured by the one-way ANOVA multiple comparison test to analyze the difference between the groups, n = 3. *$P < 0.05$, **$P < 0.01$, ***$P < 0.001$, ns, nonsignificant. **(C)** Western blots showing the relative expression of the indicated spike proteins bearing mutations from the producer cell lysates and the viral lysates. β-actin and p24 served as loading controls for cell lysates and viral lysates, respectively. **(D)** The

to the RBD-specific L452R, in conferring resistance to neutralization and promotion of infectivity.

### E156G/Δ157-158 and L452R additively effects immune escape and increased infectivity

Plasma polyclonal antibodies (from ICS-05 and ICS-03) post-recovery from the subject cases were capable of neutralizing the spike PV despite the presence of mutations (Fig S3), suggestive of the increased breadth of neutralization. For delineating key events that could have shaped the escape from vaccine-elicited polyclonal neutralizing responses, we compared the effect of combinations of the select mutants (Fig 3A) on PV neutralization by vaccine-elicited plasma polyclonal antibodies. Whereas the ability of spike E484Q and L452R to evade BNT162b2 Pfizer mRNA vaccine-elicited antibodies has been established recently (Gupta et al, 2021; Cele et al, 2021; Motozono et al, 2021), we investigated if the indicated mutations acted in synergy in the antibody evasion process. Therefore, we combined the NTD-specific change E156G/Δ157-158 with E484Q and L452R (Fig 3A) and performed the infectivity and neutralization assays. Regardless of the background (E484Q or the L452R), the NTD-specific mutation E156G/Δ157-158 increased the infectivity ~4-fold for the spike-pseudotyped lentiviral particles in HEK293T ACE2 cells (Fig 3B; *P*-value <0.0001 from one-way ANOVA). Western blotting experiments confirmed that these mutant spikes were expressed and enriched comparably in the virions (Figs 3C and S4A). It is well known that the TMPRSS2 (a serine protease) interacts with the spike protein of the SARS-CoV-2 virus and primes it for infection in host cells. To evaluate if the presence of TMPRSS2 in the target cells would impact the phenotype, we infected ACE2- and TMPRSS2-expressing A549 target cells with the indicated viruses and found a similar trend when spike harbored E156G/Δ157-158 and L452R mutations (Fig S4B). Next, we assessed if this was a manifestation of increased spike affinity with the ACE2 receptor. Accordingly, the spike PVs were pulled down using the ACE2-IgFc microbody as reported by us previously (Mishra et al, 2021). Whereas E156G/Δ157-158 mutation alone did not show a significant difference in ACE2 binding affinity, spike PV bearing E156G/Δ157-158/L452 exhibited increased affinity almost equivalent to that of ICS-05 towards hACE2 (Fig 3D). Notably, the reduced susceptibility to neutralization observed for the ICS-05/-03 spike (11-fold) was mostly explained by a combination spike mutant that harbored E156G/Δ157-158 and L452R (sevenfold less susceptible to neutralization) (Fig 3E; *P*-values of <0.01 from Wilcoxon signed-rank test). The plasma samples tested (in Figs 2D and 3E) were time interval–matched with the subject breakthrough infection cases, with all individuals receiving two doses 1-mo apart. Although the availability of more matched samples was a constraint due to the limited number of vaccinated frontline workers then, we next asked if later increased time interval (3-mo between the two doses) influenced the outcome

regardless. For this, we included the plasma from test-negative individuals who received two doses at 3-mo interval. The sensitivity profiles of the spike harboring PVs mirrored that of short-duration (1-mo) vaccinees' plasma, indicating the response was independent of a vaccine dose interval, the age/sex, the time, or the location (Fig 3F) (*P*-values of 0.001 from Wilcoxon signed-rank test).

To further understand the reduced susceptibility to neutralization as a result of NTD alterations, we explored known complex structures of the spike with antibodies. There are two major classes of antibodies; one binds with the NTD region, whereas another binds with the RBD domain of the spike protein. The structure of antibodies bound to RBD and NTD domains of the spike reveals that the neutralization escaping mutations described in this study are present at the interface of the antibody and NTD and/or RBD domain of the spike protein (Fig 3G). This observation is consistent with our neutralization assay results which show that mutations in these regions affect PV sensitivity to neutralization. Altogether, we observed that these mutations in NTD, particularly E156G/Δ157-158, cooperated with the seeding changes in the RBD, like L452R, for neutralizing antibody escape.

### Epidemiological data suggest that mutations and deletions at specific positions in NTD are positively selected

It is well known that amino acid sequence alteration in the spike has an impact on the neutralization activity of antibodies and might help the virus to escape. Our observations with the E156G/Δ157-158 mutation in reducing the PV sensitivity to neutralization are coherent. In order to extend this observation to other SARS-CoV-2 lineages (VOI/VOC) and understand if this is a prevalent feature, we checked the frequency of mutation at each amino acid position in the NTD of the spike gene from 37 strains reported by GISAID (Shu & McCauley, 2017) (Fig 4A). Strikingly, our analysis revealed that certain residues on the spike NTD showed a higher rate of mutations, including the ones at positions 156–158, suggesting that this region is a mutational hotspot. Next, to analyze whether the residues 156–158 directly interact with the neutralizing antibodies, we mapped the interaction interface from the 17 antibodies reported earlier (Chi et al, 2020; Wang et al, 2020; McCallum et al, 2021a, 2021b; Cao et al, 2021b; Liu et al, 2021b), recognizing the spike NTD that showed enrichment of 37 residues: Q14, C15, V16, N17, L18, P26, Y28, P85, N87, T109, K113, T114, Y145, H146, K147, N148, N149, K150, W152, E154, S155, E156, F157, R158, T236, R237, R246, S247, Y248, L249, T250, P251, G252, S254, S255, and S256. The residues forming antibody-recognizable epitopes in the NTD region spans from 14 to 20, a β-sheet spanning from 144 to 158, and a loop formed by 246–256 residues appears as the top interaction position in the NTD of the spike (Fig 4B and Supplemental Data 1). These three regions are thus most crucial for neutralizing antibodies to function

---

susceptibility of each spike mutant PV to neutralization by antibodies in the plasma obtained from vaccinated, test-negative individuals is plotted. Each data point represents mean $NT_{50}$ values (50% neutralization titre) obtained against the indicated virus. The $NT_{50}$ values were determined in triplicate, and geometrical means were calculated. The dotted red line represents the median response of each spike PV. The fold difference in response to the neutralizing plasma was measured compared to the reference D614G mutant spike PV (n = 6). The statistical significance was calculated by the Wilcoxon signed-rank test, two-tailed, nonparametric. Source data are available for this figure.

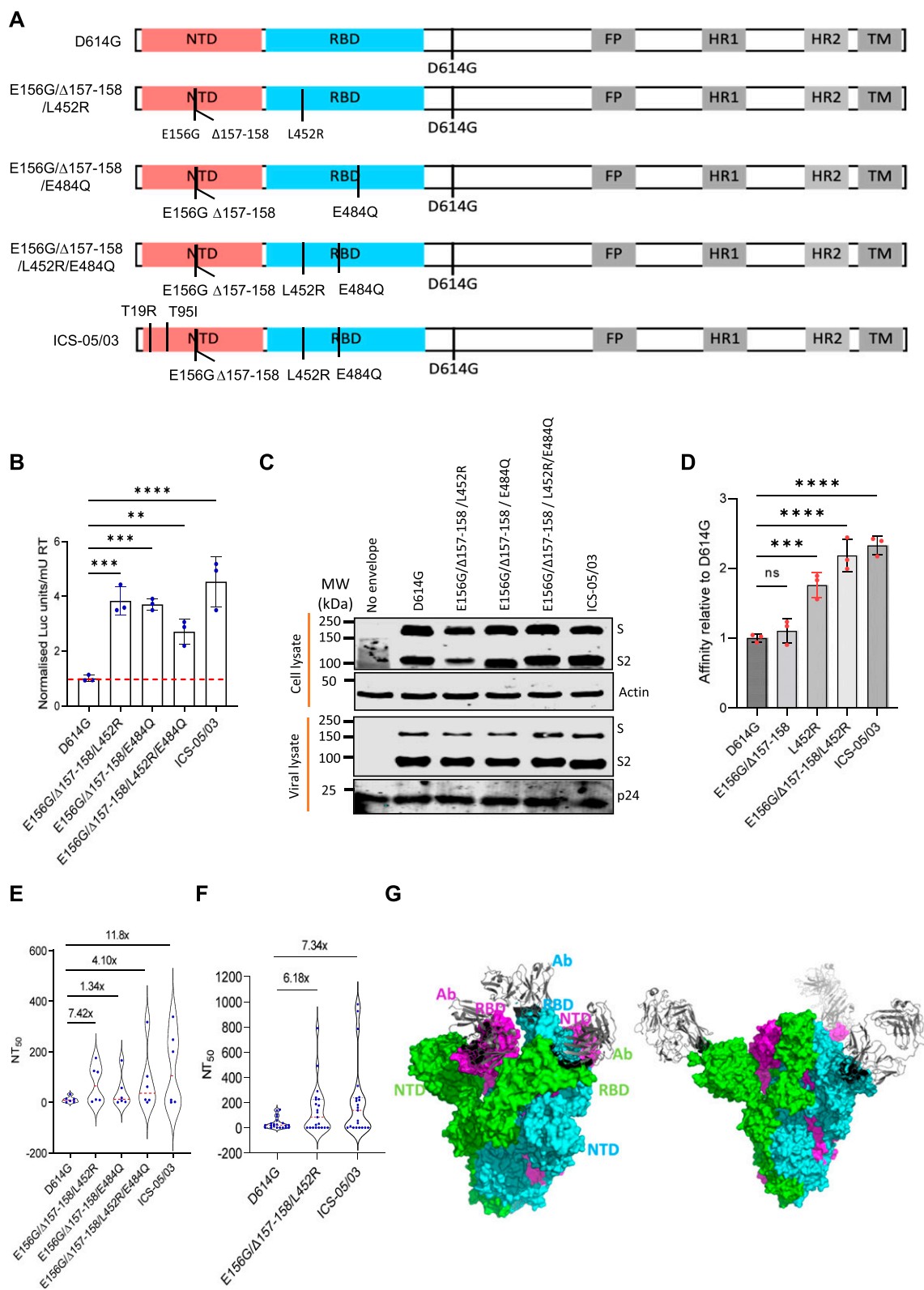

**Figure 3. Infectivity and neutralization of spike pseudoparticles and structural analysis.**
**(A)** Schematics of the spike mutants generated to check the combined effects of mutations. Amino acid positions are represented with respect to the Wuhan HU-1 sequence (NC_045512). **(B)** Infectivity profiles of the indicated spike mutant–pseudotyped lentiviruses (PV) in HEK293T ACE2 cells. The infectivity was normalized to the D614G-pseudotyped lentiviral particles. The data represent the mean of three replicates, and the significance was measured by the one-way ANOVA multiple comparison test to analyze the difference between the groups, n = 3. *P < 0·05, **P < 0·01, ***P < 0·001, ****P < 0·0001. **(C)** Western blots show the relative expression of the indicated

(Resende et al, 2021). We see that antibodies targeting the NTD of spike mostly share these residues (Fig 4B). Essentially, we found a strong correlation in the higher mutation frequency of residues present with the nAb interaction site, suggestive of a selective pressure causing a directional change in the spike protein (Table 1). These sites of higher mutation prevalence that interact with nAbs represent an NTD supersite (Fig 4C). Interestingly, the most potent NTD-targeting neutralizing antibodies (S2X333 and S2M28) earlier isolated from patients indeed form polar interactions with the NTD supersite residues (Fig 4D). The presence of the mutations in regions ranging from 144 to 158, comprising the deletion mutation E156G/Δ157-158, therefore could impair the binding by such neutralizing antibodies while maintaining the ACE affinity and help the virus escape from immune surveillance. Recurrent NTD changes in Delta and Omicron variants further supports this hypothesis (Planas et al, 2021; Cao et al, 2021a).

### Cell-to-cell fusion is enhanced by the NTD- and RBD-specific amino acid changes

The pathogenicity of SARS-CoV-2 is also attributed to the spike-dictated formation of syncytia, a phenotype characterized by cells with abnormal morphology and frequent multinucleation. Cell-to-cell fusion appears to be a more prominent feature of delta variants (Buchrieser et al, 2020; Planas et al, 2021) that can offer an avenue for evasion of humoral responses (Buchrieser et al, 2020; Bussani et al, 2020). Our analysis revealed that E156G/Δ157-158 together with L452R conferred most of the resistance to antiviral immunity elicited by vaccination. We next asked if these amino acid changes also promoted cell-to-cell fusion leading to syncytium formation. For this, we designed the assay that revealed the ability of cells expressing indicated spike mutants (Red) to fuse with bystander ACE2-expressing cells (Green). Although E156G/Δ157-158 and L452R alone had an indiscernible effect on syncytium formation compared to the reference D614G, the area after cell-to-cell fusion was increased ~2-fold (P-values of <0.0001 from one-way ANOVA) when the cells expressing D614G spike bearing L452R and the E156G/Δ157-158 mutations were cocultured with ACE2-positive cells (Fig 5A and B). Furthermore, when we combined RBD-specific L452R and E484Q with NTD-specific E156G/Δ157-158, the magnitude of the cell–cell fusion was similar to that of the ICS-05/-03 spike. Altogether, these results indicated the increased

ability of the spike from breakthrough infection cases is conferred by NTD- and RBD-specific changes that acted in concert to promote syncytium formation.

## Discussion

Little is known about the functional consequences of NTD-specific changes in the spike of emerging SARS-CoV-2 variants. In contrast, RBD-specific changes in the spike have been considered a defining feature that confers fitness to the virus. Here, we demonstrated that molecular features of the spike (such as L452R and E484Q), already known to confer fitness (Ferreira et al, 2021; Mlcochova et al, 2021; Wang et al, 2021), act in concert with alterations in the NTD to enhance spike function further. The widespread NTD-specific mutation E156G/Δ157-158, which is also shared by delta variant, likely conferred an evolutionary advantage and might underlie vaccine breakthroughs. One of Delta variant's siblings, a variant called Lambda, also carries changes in the spike NTD, in addition to L452Q, and these alterations have been associated with the virus' higher infectivity and immune evasion ability (Acevedo et al, 2021 Preprint; Kimura et al, 2021 Preprint). In agreement, we found that the ICS-05/-03 spike carried mutations in the NTD-coding sequence and that these changes indeed acted in concert in evading antiviral immunity elicited by the vaccine and contributed to increased infectivity. Furthermore, consistent with the previous reports, we also observed ACE2-expressing cells forming larger syncytia when mixed with spike-expressing cells, particularly the ICS-05/-03 spike (Asarnow et al, 2021; Planas et al, 2021; Rajah et al, 2021). It has been found that the extent of syncytia formation in SARS-CoV-2–infected patients' lungs positively correlates with disease severity and higher mortality (Buchrieser et al, 2020; Bussani et al, 2020). Our observations that E156G/Δ157-158 and L452R/E484Q mutations bearing spike induced large syncytia formation, almost equivalent to that of the ICS-05/03 spike, exemplifies NTD- and RBD-specific changes in promoting cell–cell fusion.

One of the limitations was that we did not have access to the plasma from before infection to appreciate the vaccine efficacy for the breakthrough infection cases. Thus, we elucidated the virological properties by employing the plasma samples that were time and dose interval–matched. Although we see a significant contribution of NTD-specific changes in determining resistance to

---

spike proteins bearing mutations from the producer cell lysates and the viral lysates. β-actin and p24 served as loading controls for cell lysates and viral lysates, respectively. **(D)** Pull-down of the indicated spike pseudovirus using the protein G–bound ACE2-IgFc microbody. The affinity for the ACE2 receptor was normalized to the D614G-pseudotyped lentiviral particles. The data represent the mean of three replicates, and the significance was measured by the one-way ANOVA multiple comparison test to analyze the difference between the groups, n = 3. $*P < 0.05$, $**P < 0.01$, $***P < 0.001$, $****P < 0.0001$. **(E)** The susceptibility of each spike mutant PV to neutralization by antibodies in the plasma obtained from vaccinated, test-negative individuals is plotted. Each data point represents mean $NT_{50}$ values (50% neutralization titre) obtained against the indicated virus. The $NT_{50}$ values were determined in triplicate, and means were calculated. The dotted red line represents the median response of each spike-pseudotyped virus. The fold difference in response to the neutralizing plasma was measured compared to the reference D614G mutant spike PV (n = 6). The statistical significance was calculated by Wilcoxon signed-rank test, two-tailed, nonparametric. **(F)** The susceptibility of spike mutant PV to neutralization by the antibodies in the plasma obtained from vaccinated, test-negative individuals who obtained the first and second dose at an interval of 3 mo. The dotted red line represents the median response of each spike PV. The fold difference in response to the neutralizing plasma was measured compared to the reference D614G mutant spike PV (n = 14). The statistical analysis was done by Wilcoxon signed-rank test, two-tailed, nonparametric. **(G)** The complex structure of the receptor-binding domain–specific antibody bound with a trimer of spike proteins (PDB ID: 6XEY). The surface of spike protein protomers are shown in green, cyan, and magenta, and the antibody (grey) bound to the receptor-binding domain of the spike protein is shown as a cartoon (left panel). The complex structure of the N-terminal domain–specific antibody bound with a trimer of spike proteins (PDB ID: 7C2L). The surface of spike protein protomers is shown in green, cyan, and magenta, and the antibody (grey) bound to the N-terminal domain of the spike protein is shown as a cartoon (right panel). Source data are available for this figure.

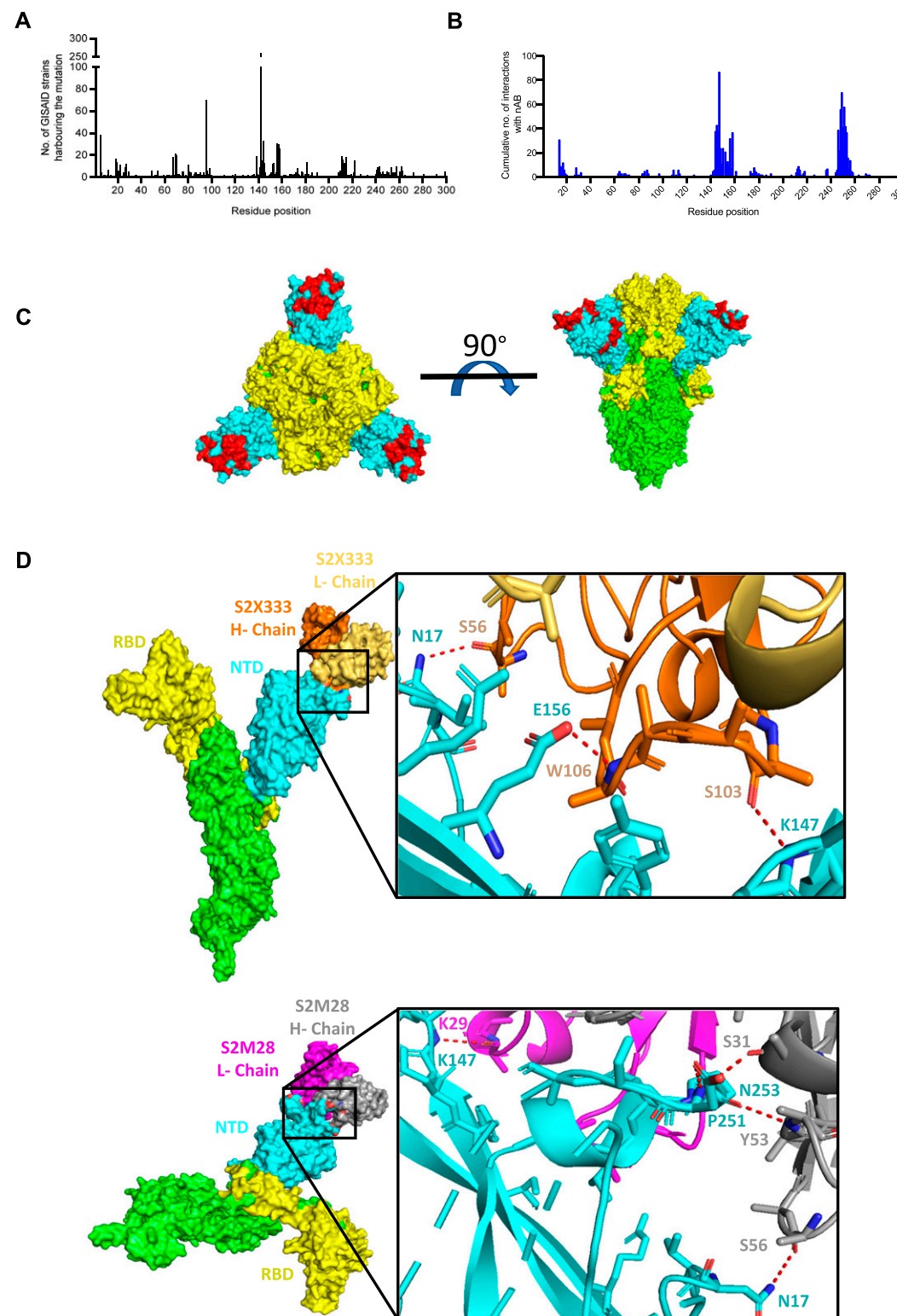

**Figure 4. Spike N-terminal domain (NTD) mutation profiles and the residues targeted by neutralizing antibodies.**
**(A)** Number of GISAID strains harboring mutations on the NTD compared to the Wuhan strain. **(B)** The cumulative number of interactions at the interface of 17-nABs and spike NTD residues predicted using the Prodigy web server. **(C)** Mapping of highly interacting residues (red) on the NTD (14–305) (cyan) of the SARS-CoV-2 spike (PDB ID: 7JJI). The receptor-binding domain (330–675) of the spike denoted in yellow. **(D)** Expanded view of S2X333 Fab (PDB ID: 7LXY) and S2M28 Fab (PDB ID: 7LY2) interactions with NTD residues. The residues at the Fab–NTD interface have been shown as sticks. Oxygen atoms in orange, nitrogen atoms in blue, and polar interactions between the interacting residues have been shown as red dashed lines.

**Table 1. Prevalence of mutations in the SARS-CoV-2 spike N-terminal domain region among various variants.**

| Mutation | Lambda | Mu | Delta | Omicron | Alpha | Beta | Gamma |
|---|---|---|---|---|---|---|---|
| L18F | 0.5 | 0.3 | 0.4 | 0.1 | 0.7 | 43.5 | 97.2 |
| T19R | 0.1 | 0.1 | 98.3 | 0.1 | 0 | 0.1 | 0.1 |
| T20N | 0.1 | 0.1 | 0 | 0 | 0.1 | 0.1 | 96.7 |
| P26S | 0.2 | 0.1 | 0.2 | 0.1 | 0.1 | 0.1 | 96.9 |
| A67V | 0.2 | 0.1 | 0.3 | 94.2 | 0.2 | 0.1 | 0.1 |
| Δ69/70 | 0.7 | 0.2 | 0.2 | 93.1 | 96.6 | 0.1 | 0.3 |
| G75V | 94 | 0.1 | 0.1 | 0.1 | 0.1 | 0.4 | 0.3 |
| T76I | 96.6 | 0.1 | 0.1 | 0.1 | 0.1 | 0.1 | 0.2 |
| D80A | 0 | 0 | 0 | 0 | 0 | 97.1 | 0.1 |
| T95I | 0.3 | 94.4 | 38.4 | 93.5 | 0.3 | 0.1 | 0.1 |
| D138Y | 0.3 | 0.1 | 0.1 | 0.1 | 0.2 | 0.6 | 94.8 |
| G142D | 0.1 | 0.1 | 66 | 95 | 0.1 | 0.1 | 0.1 |
| Δ143/145 | 0.1 | 0 | 0.1 | 91.8 | 0 | 0.1 | 0.1 |
| Δ144/144 | 0.8 | 2.1 | 0.2 | 0.1 | 95.1 | 0.7 | 0.5 |
| Y144S | 0 | 76.8 | 0 | 0 | 0 | 0 | 0 |
| Y145N | 0.1 | 85.7 | 0 | 0.1 | 0 | 0.1 | 0 |
| E156G | 0.1 | 0.1 | 92.1 | 0.1 | 0.1 | 0.1 | 0.1 |
| Δ157/158 | 0 | 0.1 | 92.2 | 0.1 | 0 | 0 | 0.1 |
| R190S | 0.1 | 0.1 | 0 | 0.1 | 0 | 0.1 | 92.9 |
| N211I | 0.1 | 0.1 | 0.1 | 83.4 | 0.1 | 0 | 0.1 |
| Δ212/212 | 0.1 | 0.1 | 0.1 | 83.8 | 0 | 0 | 0 |
| D215G | 0.1 | 0.1 | 0.1 | 0 | 0.1 | 94.6 | 0.1 |
| Δ241/243 | 0.1 | 0.1 | 0.1 | 0.1 | 0.1 | 89.5 | 0.1 |
| R246N | 81.7 | 0 | 0 | 0 | 0 | 0.1 | 0.1 |
| Δ247/253 | 83.4 | 0 | 0 | 0 | 0 | 0 | 0.1 |

neutralization for time and dose interval–matched samples, it was also consistent irrespective of the time of vaccine administration and the dosing interval. The variability observed in the neutralization profiles using PVs, however, was expected as it basically reflects differential antibody responses between various individuals.

The growing evidence that NTD-specific mutations are modulating antigenicity and response to vaccine-elicited immunity requires immediate attention. Only a few NTD-targeting mAbs have been structurally characterized (Chi et al, 2020; Suryadevara et al, 2021). Recent studies have linked the spike NTD mutations with virus transmission and escape from neutralizing antibodies (Meng et al, 2021; Suryadevara et al, 2021). As observed from the sequencing data on GISAID, several sequences belonging to emerging variants are found with a deletion/mutation in the NTD region of the spike. Beta variant spike proteins carry three amino acid deletion LAL242-244 in the NTD (Meng et al, 2021; Wang et al, 2021), which does not change the neutralization effect of antibodies, but specific antibodies such as 4A8 are shown not to neutralize the virus (Wang et al, 2021; Wibmer et al, 2021). Also, the Lambda variant possessing the RSYLTPGD246-253N mutation in the spike NTD has been shown to confer resistance to vaccine-elicited

antibodies, particularly antibodies targeting the NTD "supersite" (Gabriele et al, 2021; M et al, 2021; Suryadevara et al, 2021; Harvey et al, 2021; Kimura et al, 2021 *Preprint*). The NTD-specific changes that we reported here are closely associated with breakthrough infection cases. Its prevalence worldwide indicates the selective pressure to maintain this change for higher infectivity and immune escape.

Most recently, a new SARS-CoV-2 variant Omicron (B.1.1.159) is identified. The Omicron was first detected in Botswana (11 November 2021) and now has spread across the globe, targeting vaccinated individuals (Wang et al, 2021). Omicron harbors multiple mutations (46 mutations genome-wide), especially in the spike protein, which underlies its high transmissibility and infectivity (Saxena et al, 2022). The mutation profile of the Omicron spike shows recurrent deletion in the NTD which coincides with our finding (deletion in the 142–158 and 246–256 loops) and plausibly underlies its ability to escape vaccine-elicited immunity. Whereas the extent to which spike mutations will enable variants to evade vaccine-elicited or natural immunity remains to be determined, the role of the NTD in determining infectivity and immune escape appears to be an emerging hallmark.

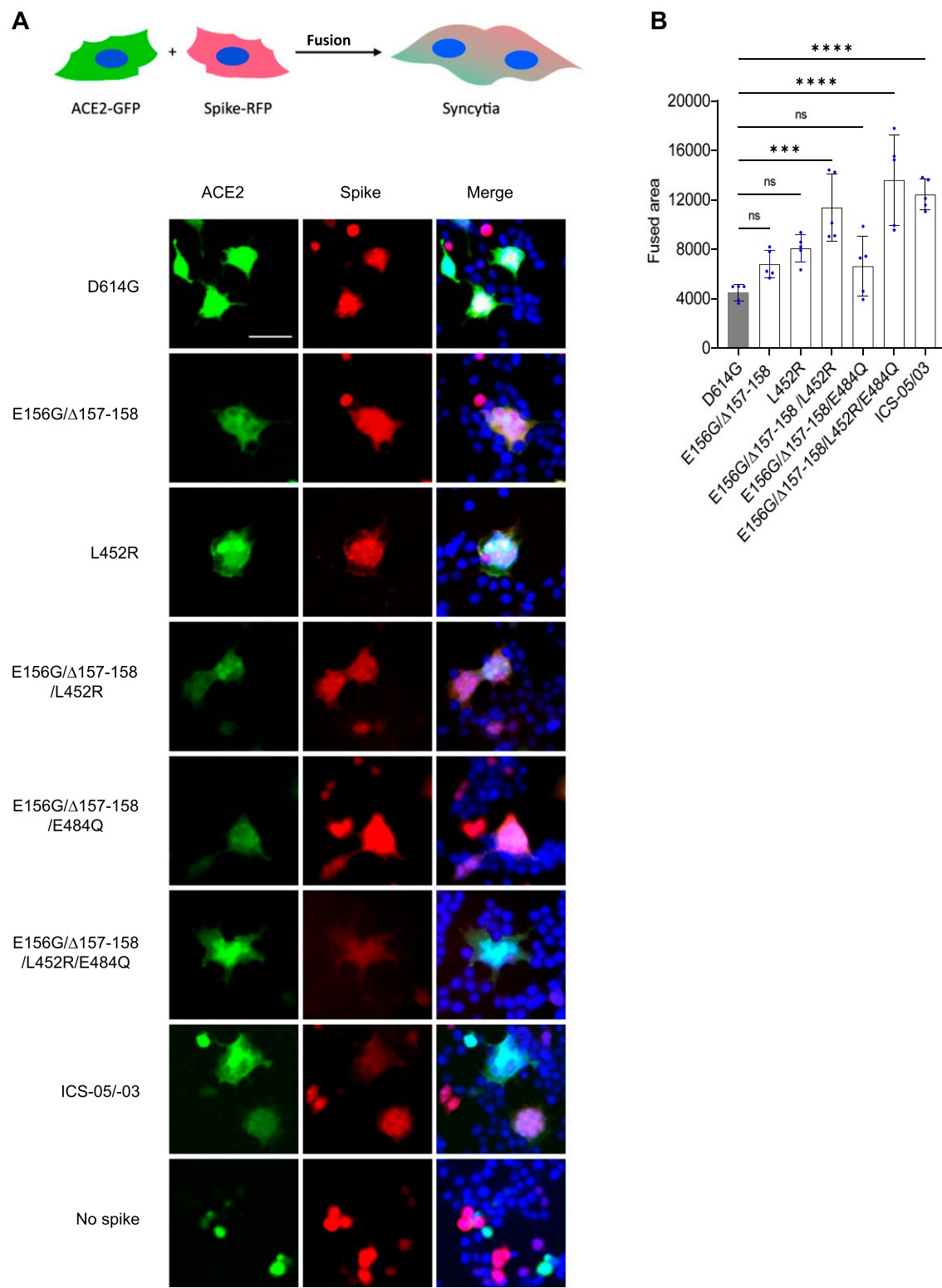

**Figure 5. SARS-CoV-2 spike mutants form syncytia in ACE2-expressing cells.**
**(A)** The HEK293T ACE2-transfected cells (green) were mixed with indicated spike mutants transfected cells (red) displaying syncytia formation. The cells with ACE2 expression mixed with the control cells with no spike expression served as the control. Scale 50 $\mu$m. **(B)** Quantification of the cell–cell fusion by measuring the fused area for five different fields from three replicates. The bar graph represents mean ± SD. Statistical significance was calculated by one-way ANOVA.

# Materials and Methods

## Ethics statement

The Institute Ethics Committee approved this study. All the samples were collected with due consent from the donors.

## Plasmids

The plasmid expressing SARS-CoV-2 spike protein (Wuhan isolate) was procured from Addgene with 19–amino acid deletion at the C-terminal that enables efficient lentiviral packaging. The pcDNA 3.1 bs (-) spike D614G mutant was generated by site-directed mutagenesis first (Mishra et al, 2021). All other plasmids expressing spike protein mutants such as pcDNA 3.1 bs(-) T19R, pcDNA 3.1 bs(-) T95I, pcDNA 3.1 bs(-) E156G/Δ157-158, pcDNA 3.1 bs(-) L452R, pcDNA 3.1 bs(-) E484Q, pcDNA 3.1 bs(-) E156G/Δ157-158/L452R, pcDNA 3.1 bs(-) E156G/Δ157-158/E484Q, pcDNA 3.1 bs(-) E156G/Δ157-158/L452R/E484Q, and pcDNA 3.1 bs(-) ICS-05 were generated using site-directed mutagenesis by PCR using the pcDNA 3.1 bs(-) spike D614G plasmid as the template. The following primers S: D614G forward; 5'GTGCTGTACCAGGGCGTGAATTGCACC-3' reverse; 5'GGTGCAATTCACGCCCTGGTACAGCAC-3'; S: T19R forward; 5'TCTGGTCTCGTCTCAGTGCGTGAACCTGAGAACTAGAACCCAGCTGCCTC-3' reverse; 5'CTAGCAGCAGCTGCCGCAGGA-3'; S: T95I forward; 5'GCGTGTACTTCGCCTCCATTGAGAAGAGCAACATCATC-3' reverse; 5'GATGATGTTGCTCTTCTCAATGGAGGCGAAGTACACGC-3'; S: E156G/Δ157-158 forward; 5'AAGGTGCAATTGTTGGCGGAGCTGTACACGCCGCTCTCCATCCAGGACT-3' reverse; 5'AGTCCTGGATGGAGAGCGGCGTGTACAGCTCCGCCAACAATTGCACCTT3'; S: L452R forward; 5'GCAACTACAATTACCGGTACCGCCTGTTCCG-3' reverse 5'CGGAACAGGCGGTACCGGTAATTGTAGTTGC-3'; S: E484Q forward; 5'CCATGCAATGGAGTGCAGGGCTTCAACTGCT reverse; 5'AGCAGTTGAAGCCCTGCACTCCATTGCATGG-3' were used to generate the abovementioned plasmids. We use the term mutation to indicate an amino acid change with respect to the Wuhan Hu-1 reference sequence (NC_045512). All the constructs were sequence verified for the reported mutations. A list of plasmids and the relevant information are tabulated in Table S1.

## Cell culture and reagents

HEK293T cells (ECACC) were grown in the DMEM medium supplemented with 10% FBS, 2 mM glutamine, and 1% penicillin–streptomycin. ACE2+ cells were generated by lentiviral transduction of HEK293T cells and were selected on the hygromycin. Reagent details are provided in Table S2.

## Pseudovirus production

HEK293T cells ($3 \times 10^6$ cells) were seeded in a 10-$cm^2$ plate 24 h before transfection for the spike-pseudotyped lentivirus production. The cells were co-transfected, with pScalps Zsgreen Luciferase (8 $\mu$g), psPAX2 (6 $\mu$g), pcDNA 3.1 bs(-) N protein-encoding plasmid (2 $\mu$g), and either 2 $\mu$g of the parental spike (D614G) or its derivatives plasmids, by the calcium phosphate transfection method (Mishra et al, 2021). The cells were replenished with a fresh medium after 16 h of transfection. The supernatant containing the viral particles was collected 48 h post-transfection. The supernatant was centrifuged at 300$g$ for 5 min and passed through a 0.22-$\mu$m filter to remove cell debris. The viruses were quantified using an SGPERT assay to normalize the input and loading (Pizzato et al, 2009). To check the virion incorporation of the spike protein, the supernatant containing viral particles was overlaid on top of a sucrose (20%) cushion and centrifuged at 100,000$g$ for 2 h at 4°C. After centrifugation, the supernatant was removed completely, and the virus pellet was resuspended in Laemmli buffer containing 10 mM TCEP as a reducing agent.

## Transduction and infectivity measurement

All the infectivity experiments were performed in 96-well plate (Eppendorf) formats with 40–60% target cell confluency. For infectivity experiments, HEK293T ACE2+ cells were seeded 24 h before transduction. The transduction was done in quadruplicates with dilutions (undiluted, 1:5, 1:25, 1:125) of pseudotyped virus preparation as described earlier (Mishra et al, 2021). The cells incubated with a growth medium with heat-inactivated FBS alone were considered as the control. The level of transduction was quantified using luciferase assay.

## Luciferase assay

The luciferase assay was performed to quantify the level of transduction by spike-variant lentiviruses. For this, the growth medium was removed from each well, and cells were washed with 1× PBS. Further, 100 $\mu$l of lysis buffer was added to each well to lyse transduced cells at room temperature for 20 min. Fifty microliters of lysate was transferred to a white 96-well plate and finally mixed with the 50 $\mu$l of the substrate solution, and enzyme activity was measured using SpectraMax i3X (Molecular Devices).

## Collection of the plasma and pseudovirus neutralization

For obtaining plasma, 5 ml of venous blood sample was collected in EDTA-containing vials and mixed gently by inverting multiple times to avoid coagulation. Furthermore, the tubes were centrifuged at 200$g$ for 15 min at 4°C in a swing-out bucket. The plasma layer above the PBMCs was collected as a source of antibodies. The plasma of the individual, who got two doses of Covishield and became COVID-19 positive, was collected after the RT-PCR–negative report post-recovery. The plasma from test-negative vaccinated individuals who got both doses of the Covishield vaccine was collected after 15–30 d of vaccination. The plasma of total 20 vaccinees (average age: 29 years, range: 22–42, 65% male and 35% females) were inactivated at 56°C for 30 min and stored at −80°C until use.

To check the neutralization titre ($NT_{50}$) potential of antibodies present in the plasma of vaccinated (Covishield) and a vaccinated individual post–COVID-19 recovery, various dilutions of plasma (10, 1, $10^{-1}$, $10^{-2}$, $10^{-3}$, $10^{-4}$ $10^{-5}$, and $10^{-6}$) were incubated with equivalent spike PV for 20 min at room temperature before challenging the HEK293T ACE2+ cells. The amount of virus entering the target cells was measured using luciferase units after 48 h of transduction. To

obtain the $NT_{50}$ value, a neutralization curve and the analysis output was generated using GraphPad Prism 9.

### Affinity assessment of spike variants using the ACE2-IgFc microbody

To estimate the binding affinity of spike mutants for the hACE2 receptor, various spike-pseudotyped viruses were produced from HEK293T cells. Next, the RT units were estimated using SGPERT assay. In parallel, 50 $\mu$l of protein G magnetic beads were washed twice with 10% FBS–containing DMEM medium, and then 100 $\mu$l (50 $\mu$g) of the ACE2-IgFc microbody was added to protein G beads and incubated for 20 min with mixing in between. After 20 min, the beads were placed on a magnetic rack and washed thrice with 10% FBS-containing DMEM medium to remove the unbound ACE2-IgFc microbody. Further beads were resuspended in 50 $\mu$l DMEM medium and split in five microcentrifuge tubes containing 10 $\mu$l of beads each. Next, equal number of various spike PV was added to the beads and incubated for 10 min at room temperature. The microcentrifuge tubes were placed back on the magnetic rack, and beads were washed thrice with 1× core buffer. In the end, the bead-bound viral particles were lysed using 20 $\mu$l of 2× lysis buffer. After lysis, the 1× core buffer was added to make its volume up to 200 $\mu$l. The 10 $\mu$l of this was used to quantify the viral particles using SGPERT assay.

### Cell-to-cell fusion assay

For studying fusion of ACE2-expressing cells with spike-expressing cells, we seeded HEK293T cells in 24-well plates at 70–80% confluency and co-transfected separately each well with Tag-RFP 657 (50 ng) along with spike protein–expressing vector (500 ng) harboring indicated mutations. Separately, HEK293T cells were co-transfected with pEGFP-N1- (50 ng) and ACE2-expressing vectors (500 ng). After 10 h of transfection, cells were trypsinized, mixed at 1:1 ratio (spike: ACE2), and seeded in 96-well plates. After 48 h of transfection, cells were counterstained with Hoechst. Cells were fixed with 4% paraformaldehyde for 20 min at room temperature and washed thrice with PBS to remove PFA and taken for imaging using the Thermo Scientific CellInsight CX7 High-Content Screening (HCS) Platform. To calculate the effect of spike mutations on syncytium formation, five different fields were randomly chosen, and the area of fused cells was measured using ImageJ software.

### Western blotting

The expression of spike protein mutants was checked from the virus-producing HEK293T cells. After 48 h of transfection, cells were collected and lysed in RIPA buffer supplemented with 2× PIC (protease inhibitor cocktail) and 50 mM TCEP at 4°C for 30 min. Furthermore, the supernatant was collected after centrifugation at 17,000$g$ for 10 min at 4°C. The viral particle lysate and cell lysate were run on the 8% Tris–Tricine PAGE gel following mixing lysates with 4× Laemmli. Thereafter, proteins were transferred onto the Polyvinylidene fluoride (PVDF) membrane (Immobilon-FL; Merck-Millipore). After the electrotransfer, the membrane was blocked with the membrane blocking reagents (Sigma-Aldrich) followed

by primary and secondary antibody incubations for 1 h at room temperature, each of which was followed by three washes with TBST. For the p24 and $\beta$-actin detection, anti-p24 (NIH ARP), rabbit anti-$\beta$ actin (Cat. no. 926-42210, RRID:AB_1850027; LI-COR Biosciences), and for SARS-CoV-2 spike detection, mouse anti-spike (Cat. no. ZMS1076; Sigma-Aldrich), respectively, were used as primary antibodies. The IR dye 680 goat anti-mouse was used as a secondary antibody for the anti-spike antibody. The IR dye 800 goat anti-mouse or IR dye 800 goat anti-rabbit (Cat. no. 925–68070, RRID: AB_2651128; LI-COR Biosciences, and Cat. no. 925–32211, RRID: AB_2651127; LI-COR Biosciences) were used against primary p24 and $\beta$-actin antibodies, respectively.

### Structural analysis of the spike with NTD-specific neutralizing antibodies

All the crystal structures of the spike protein and NTD-specific antibodies were obtained from Protein Data Bank (PDB). The interacting residues of spike NTD and nAb were predicted using the PRODIGY web server (Vangone & Bonvin, 2015). Furthermore, all polar interactions were validated using PyMOL (PyMOL, 2017). The PDB IDs for all the structures used in the studies are provided in the Supplemental Data 1.

### Sequence data analysis

To interpret the frequency of insertion and deletion in the SARS-CoV-2 S gene, sequences were retrieved from GISAID data (Shu & McCauley, 2017). Sequences having deletion (which causes frameshift mutation) were discarded from the analysis. The sequences were classified into SARS-CoV-2 strains and were analyzed for frequency of insertion and deletion at each amino acid in the NTD of spike sequence using Nextrain (https://nextstrain.org/).

### Software and statistical analysis

All graphs were generated using GraphPad Prism (version 9·0). Statistical analysis was carried out using the in-built algorithms bundled with the software. Specific portions of images were produced using BioRender. PyMOL (version 2) and AlphaFold were used for protein structure visualization. Western blot images were processed using Image Studio Lite Ver. 5·2 (LI-COR Biosciences). ImageJ was used for image processing. The images captured using the CX7 High-Content Screening Platform were analyzed using the Thermo Scientific HCS Studio.

## Data Availability

All data and materials used in the analyses are included in the manuscript.

## Supplementary Information

# Acknowledgements

The authors thank Siva Umapathy, Nevan Krogan, Massimo Pizzato, Krishna Jain, Raffaele De Francesco, Didier Trono, and the NIH AIDS Reagent Program for helpful discussions and various reagents. Vipin Bhardwaj and Pavitra Ramdas are acknowledged for technical help. This work was supported by intramural funds from the IISER Bhopal. A Chande is a recipient of the DBT/Wellcome Trust India Alliance Fellowship (grant number IA/I/18/2/504006). A fellowship from the MHRD supports T Mishra and G Joshi. R Dalavi is supported by a fellowship from CSIR.

## Author Contributions

T Mishra: data curation, software, formal analysis, validation, investigation, visualization, methodology, and writing—original draft, review, and editing.
R Dalavi: data curation, software, formal analysis, validation, investigation, visualization, methodology, and writing—review and editing.
G Joshi: formal analysis, validation, investigation, visualization, methodology, and writing—review and editing.
A Kumar: Resources, data curation, software, formal analysis, investigation, visualization, methodology, and writing—review and editing.
P Pandey: investigation, visualization, methodology, and writing—review and editing.
S Shukla: resources, data curation, formal analysis, validation, investigation, visualization, methodology, and writing—review and editing.
RK Mishra: resources, data curation, formal analysis, validation, investigation, visualization, methodology, and writing—review and editing.
A Chande: conceptualization, resources, data curation, software, formal analysis, supervision, funding acquisition, validation, investigation, visualization, methodology, project administration, and writing—original draft, review, and editing.

## Conflict of Interest Statement

The authors declare that they have no conflict of interest.

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
