## [Reviewer comments · Life Science Alliance]

Life Science Alliance

SARS CoV-2 spike E156G/ Δ 157-158 mutations contribute to increased infectivity and immune escape

Tarun Mishra, Rishikesh Dalavi, Garima Joshi, Atul Kumar, Pankaj Pandey, Sanjeev Shukla, Ram Mishra, and Ajit Chande
DOI: <https://doi.org/10.26508/lsa.202201415>

Corresponding author(s): Ajit Chande, Indian Institute of Science Education and Research, Bhopal

Review Timeline:	Submission Date:	2022-02-17
	Editorial Decision:	2022-02-28
	Revision Received:	2022-03-01
	Accepted:	2022-03-02

Scientific Editor: Novella Guidi

Transaction Report:

Please note that the manuscript was reviewed at Review Commons and these reports were taken into account in the decision-making process at Life Science Alliance.

Review
COMMONS

February 28, 2022

RE: Life Science Alliance Manuscript #LSA-2022-01415

Dr. Ajit Chande
Indian Institute of Science Education and Research, Bhopal
Biological Sciences
Unknown
India [IN]

Dear Dr. Chande,

Thank you for submitting your revised manuscript entitled "The role of SARS CoV-2 spike NTD mutations in determining neutralization susceptibility and virion infectivity". We would be happy to publish your paper in Life Science Alliance pending final revisions necessary to meet our formatting guidelines.

- Please upload all figure files as individual ones, including the supplementary figure files; all figure legends should only appear in the main manuscript file
- please add a Running Title to our system
- please add a Summary Blurb/Alternate Abstract and Category for your manuscript in our system
- please add the Twitter handle of your host institute/organization as well as your own or/and one of the authors in our system
- please add an Author Contributions section to your main manuscript text
- please use Capital letters when introducing panels in the figures, their legends, and callouts in the manuscript text
- please add your main, supplementary figure, and table legends to the main manuscript text after the references section
- Please indicate molecular weight next to each protein blot

A. FINAL FILES:

B. MANUSCRIPT ORGANIZATION AND FORMATTING:

Sincerely,

March 2, 2022

RE: Life Science Alliance Manuscript #LSA-2022-01415R

Dr. Ajit Chande
Indian Institute of Science Education and Research, Bhopal
Biological Sciences
Unknown
India [IN]

Dear Dr. Chande,

Thank you for submitting your Research Article entitled "SARS CoV-2 spike E156G/Δ157-158 mutations contribute to increased infectivity and immune escape". It is a pleasure to let you know that your manuscript is now accepted for publication in Life Science Alliance. Congratulations on this interesting work.

DISTRIBUTION OF MATERIALS:

Again, congratulations on a very nice paper. I hope you found the review process to be constructive and are pleased with how the manuscript was handled editorially. We look forward to future exciting submissions from your lab.

Sincerely,
